# Metaepigenomic analysis reveals the unexplored diversity of DNA methylation in an environmental prokaryotic community

Satoshi Hiraoka[1,2], Yusuke Okazaki[3], Mizue Anda[4], Atsushi Toyoda [5], Shin-ichi Nakano[3] & Wataru Iwasaki [1,4,6]

DNA methylation plays important roles in prokaryotes, and their genomic landscapes—prokaryotic epigenomes—have recently begun to be disclosed. However, our knowledge of prokaryotic methylation systems is focused on those of culturable microbes, which are rare in nature. Here, we used single-molecule real-time and circular consensus sequencing techniques to reveal the 'metaepigenomes' of a microbial community in the largest lake in Japan, Lake Biwa. We reconstructed 19 draft genomes from diverse bacterial and archaeal groups, most of which are yet to be cultured. The analysis of DNA chemical modifications in those genomes revealed 22 methylated motifs, nine of which were novel. We identified methyltransferase genes likely responsible for methylation of the novel motifs, and confirmed the catalytic specificities of four of them via transformation experiments using synthetic genes. Our study highlights metaepigenomics as a powerful approach for identification of the vast unexplored variety of prokaryotic DNA methylation systems in nature.

[1] Department of Computational Biology and Medical Sciences, Graduate School of Frontier Sciences, The University of Tokyo, Kashiwa 277-8568, Japan. [2] Research and Development Center for Marine Biosciences, Japan Agency for Marine-Earth Science and Technology (JAMSTEC), Yokosuka 237-0061, Japan. [3] Center for Ecological Research, Kyoto University, Otsu 520-2113, Japan. [4] Department of Biological Sciences, Graduate School of Science, The University of Tokyo, Tokyo 113-0033, Japan. [5] National Institute of Genetics, Mishima 411-8540, Japan. [6] Atmosphere and Ocean Research Institute, The University of Tokyo, Kashiwa 277-8564, Japan. Correspondence and requests for materials should be addressed to S.H. (email: hiraokas@jamstec.go.jp) or to W.I. (email: iwasaki@bs.s.u-tokyo.ac.jp)

DNA methylation is a major class of epigenetic modification that is found in diverse prokaryotes, in addition to eukaryotes[1]. For example, prokaryotic DNA methylation by sequence-specific restriction-modification (RM) systems that protect host cells from invasion by phages or extracellular DNA has been well characterized and is utilized as a key tool in biotechnology[2–4]. In addition, recent studies have revealed that prokaryotic DNA methylation plays additional roles, performing various biological functions, including regulation of gene expression, mismatch DNA repair, and cell cycle functions[5–9]. Research interest in the diversity of prokaryotic methylation systems is therefore growing due to their importance in microbial physiology, genetics, evolution, and disease pathogenicity[7,10]. However, our knowledge of the diversity of prokaryotic methylation systems has been severely limited thus far because most studies focus only on the rare prokaryotes that are cultivable in laboratories.

The recent development of single-molecule real-time (SMRT) sequencing technology provides us with another tool for observing DNA methylation. An array of DNA methylomes of cultivable prokaryotic strains, including N6-methyladenine (m6A), 5-methylcytosine (m5C), and N4-methylcytosine (m4C) modifications, have been revealed by this technology[11–14]. Despite its high rates of base-calling and modification detection errors per raw read[15,16], SMRT sequencing technology can produce ultralong reads of up to 60 kbp with few context-specific biases (e.g., GC bias)[17]. This characteristic enables SMRT sequencing to achieve high accuracy by merging data from many erroneous raw reads originating from clonal DNA molecules, typically from cultivated prokaryotic populations[18]. Alternatively, in an approach referred to as circular consensus sequencing (CCS), a circular DNA library is prepared as a sequence template to allow the generation of a single ultralong raw read containing multiple sequences ('subreads') that correspond to the same stretch on the template[19,20]; therefore, a cultivated clonal population is not required[21]. However, CCS has thus far been applied in only a few shotgun metagenomics studies[22] and, to the best of our knowledge, has not yet been applied to 'metaepigenomics' or direct methylome analysis of environmental microbial communities, which are usually constituted by uncultured prokaryotes.

Here, we applied CCS to shotgun metagenomic and metaepigenomic analyses of freshwater microbial communities in Lake Biwa, the largest lake in Japan, to reveal the genomic and epigenomic characteristics of the environmental microbial communities using the PacBio Sequel platform (Supplementary Fig. 1a). Freshwater lakes are of economical and social importance, where microbes constitute the bases of their ecosystems[23]. In addition, freshwater habitats are rich in phage–prokaryote interactions[24–27], which can affect prokaryotic DNA methylation. We report that our CCS analyses of the environmental microbial samples allowed reconstruction of draft genomes and the identification of their methylated motifs, at least nine of which were novel. Furthermore, we computationally predicted and experimentally confirmed four methyltransferases (MTases) responsible for the detected methylated motifs. Importantly, two of the four MTases were revealed to recognize novel motif sequences.

## Results and Discussion

**Water sampling, SMRT sequencing, and circular consensus analysis.** Water samples were collected at a pelagic site in Lake Biwa, Japan, at 5 m (biwa_5m) and 65 m depths (biwa_65m), from which PacBio Sequel produced a total of 2.6 million (9.6 Gbp) and 2.0 million (6.4 Gbp) subreads, respectively (Table 1). The circular consensus analysis produced 168,599 and 117,802 CCS reads, with lengths of 4474 ± 931 and 4394 ± 587 bp,

**Table 1 Statistics of SMRT sequencing and CCS-read analysis**

| Sample | biwa_5m | biwa_65m |
|---|---|---|
| Sequenced reads | 850,494 | 688,436 |
| Total base pairs (bp) | 9,570,723,004 | 6,419,717,083 |
| CCS reads | 168,599 | 117,802 |
| Read length (bp) | 4474 ± 931 | 4394 ± 587 |
| Total base (bp) | 754,416,328 | 517,663,806 |
| 16S rRNA | 170 | 106 |
| Length (bp) | 1491 ± 64 | 1468 ± 104 |

respectively (Table 1 and Supplementary Fig. 2). In the shallow sample data, at least 90% of the CCS reads showed high quality (Phred quality scores > 20) at each base position, except for the 5′-terminal five bases and 3′-terminal bases after the 5638th base. In the deep sample data, the same was true, except for the 5′-terminal four bases and 3′-terminal bases after the 5356th base (Supplementary Fig. 3).

**Taxonomic analysis.** Taxonomic assignment of the CCS reads was performed using Kaiju[28] and the National Center for Biotechnology Information non-redundant (NCBI nr) database[29] (Fig. 1). The assignment ratios were >88% and >56% at the phylum and genus levels, respectively, which were higher than those for the Illumina-based shotgun metagenomic analysis of lake freshwater and other environments using the same computational method[28]. Kraken[30] with complete prokaryotic and viral genomes in RefSeq[31] (Supplementary Fig. 4a–c) provided similar results but resulted in much lower assignment ratios (30% and 27%, respectively), likely due to the lack of genomic data for freshwater microbes in RefSeq. The 16S ribosomal RNA (rRNA) sequence-based taxonomic assignment via blastn searches against the SILVA database[32] also provided consistent results (Supplementary Fig. 4d–f). It should be noted that 16S rRNA-based and CDS-based taxonomic assignments can be affected by 16S rRNA gene copy numbers and genome sizes, respectively.

At the phylum level, Proteobacteria dominated both samples, followed by Actinobacteria, Verrucomicrobia, and Bacteroidetes (Fig. 1). Chloroflexi and Thaumarchaeota were especially abundant in the deep water sample, consistent with previous findings[33,34]. The ratio of Archaea was particularly low in the shallow sample (0.6 and 6.9% in biwa_5m and biwa_65m, respectively). Although the filter pore-size range (5–0.2 μm) was not suitable for most viruses and eukaryotic cells, non-negligible ratios corresponding to their existence were observed in the shallow sample. The dominant eukaryotic phylum was Opisthokonta (2.68 and 0.92%), followed by Alveolata (1.67 and 0.45%) and Stramenopiles (1.45 and 0.15%). Among viruses, Caudovirales and Phycodnaviridae were the most abundant families in both samples. Caudovirales are known to act as bacteriophages, while Phycodnaviridae primarily infect eukaryotic algae. The third most abundant viral family was Mimiviridae, whose members are also known as 'Megavirales' due to their large genome size (0.6–1.3 Mbp)[35,36]. Viruses without double-stranded DNA (i.e., single-stranded DNA and RNA viruses) were not observed because of the experimental method employed. Overall, the taxonomic composition was consistent with those obtained in previous studies on microbial communities in freshwater lake environments, reflecting the fact that SMRT sequencing provides taxonomic compositions consistent with those obtained using short-read technologies, such as the Illumina MiSeq and HiSeq platforms[37,38].

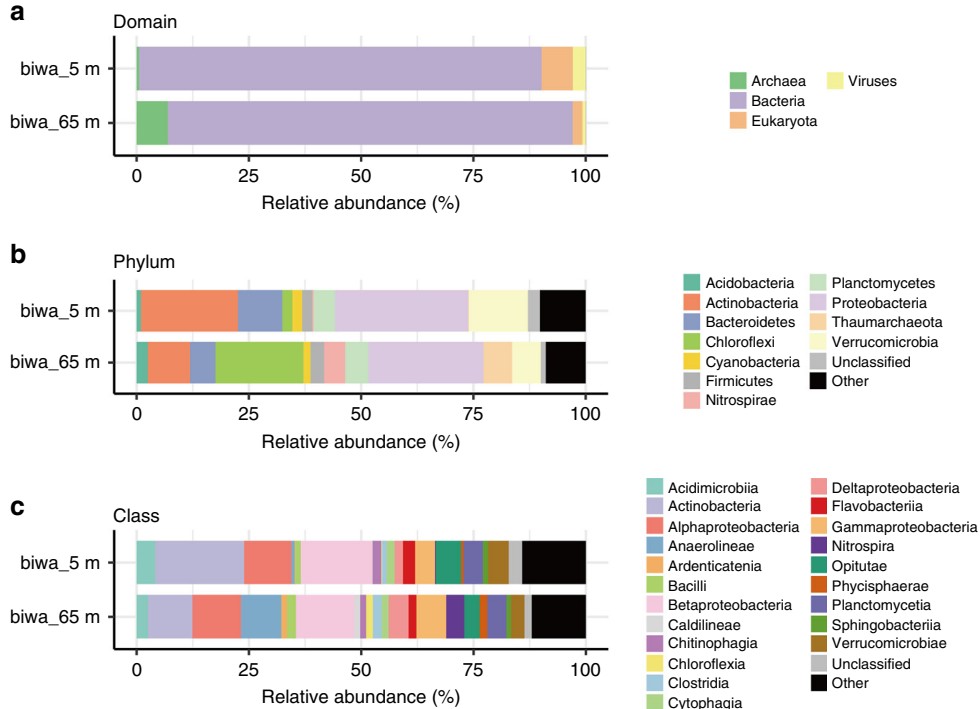

**Fig. 1** Phylogenetic distribution of CCS reads. Estimated relative abundances at the **a** domain, **b** phylum, and **c** class levels are shown. Eukaryotic and viral reads are ignored, and groups with <1% abundance are grouped as 'Other' in **b**, **c**

**Metagenomic assembly and genome binning**. The CCS reads from the shallow and deep samples were assembled into 599 and 429 contigs, respectively, using Canu[18]. After removing 45 (7.5%) and 84 (19.6%) repetitive contigs, we retrieved 554 and 345 contigs, respectively (Supplementary Table 3). The corresponding N50 values were 83 and 76 kbp, and the longest contigs had lengths of 481 and 740 kbp, respectively. Notably, the contigs were much longer than those obtained in a previous study that applied CCS for shotgun metagenomics analysis of an active sludge microbial community[22]. We also used Mira[39] for metagenomic assembly, but this resulted in shorter longest contigs (148 and 151 kbp, respectively) and N50 values (19 and 18 kbp, respectively).

The contigs were binned to genomes using MetaBAT[40], which is a reference-independent binning tool, based on CCS-read coverage and tetranucleotide frequency (Fig. 2 and Table 2). Among a total of 554 and 345 contigs, 290 (52.3%) and 100 (29.0%) were assigned to 15 and 4 bins from the shallow and deep samples, respectively. In total, 46.9 and 44.8% of the CCS reads could be mapped to the draft genomes for the shallow and deep samples, respectively. We obtained a draft genome for each bin, where the completeness of the genome ranged from 17 to 99% (67% on average). Estimated contamination levels were low (<3% in each draft genome). Based on the total contig size and estimated genome completeness of each draft genome, the genome sizes were estimated to range from 1.0 to 5.6 Mbp. The GC content ranged from 29 to 68%, and the N50 was 24 kbp on average, with a maximum of 1.67 Mbp.

The 19 draft genomes belonged to 7 phyla (Table 2 and Supplementary Fig. 5). Among these draft genomes, 10 contained 16S rRNA genes, and many of them showed top hits to uncultured clades; thus, our CCS-based approach was estimated to have truly targeted multiple uncultured prokaryotes. Seven draft genomes were predicted to belong to the phylum

Actinobacteria, including *Candidatus* Planktophila (BS7), one of the most dominant bacterioplankton lineages in freshwater systems[23,41]. The draft genomes affiliated with other dominant freshwater lineages were also recovered, including *Candidatus* Methylopumilus (BS12)[42], the freshwater lineage (LD12) of Pelagibacterales (BS14)[43,44], and Nitrospirae (BD2) and *Candidatus* Nitrosoarchaeum (BD3), the predominant nitrifying bacteria and archaea in the hypolimnion[33,34]. Four draft genomes were affiliated with the phylum Verrucomicrobia (BS6, BS8, BS10, and BD4), in line with a previous study[45]. The BS3 and BD1 draft genomes likely represent members of the CL500-11 group (class Anaerolineae) of the Chloroflexi phylum, where BD1 presented the highest coverage of >45×. This group is a dominant group in the hypolimnion of Lake Biwa and is frequently found in deep oligotrophic freshwater environments worldwide[46]. Although Proteobacteria is the most dominated phylum, two and no draft genomes were retrieved from the shallow and deep samples, respectively. Regarding the shallow sample, approximately one-fourth of the Proteobacteria CCS reads could be mapped to the two draft genomes, which means three-fourths of them likely originated from minor and diverse Proteobacteria clades. Overall, the phylogeny of the reconstructed genomes likely reflects the major lineages that are yet to be cultured but are dominantly present in the water of Lake Biwa.

**Metaepigenomic analysis**. A total of 29 candidate methylated motifs were detected in 10 draft genomes (Table 3). Their methylation ratios ranged from 19 to 99%, which can be affected by modification detection power, i.e., these ratios are likely lower than the true methylation levels. The mapped subread coverages of the methylated motifs ranged from 28.7 to 297.3×. Three motifs from the Proteobacteria BS12 genome contained similar sequences (HCAG**C**TKC, BGMAG**C**TGD, and GMAG**C**TKC, where B: C/G/T, D: A/G/T, H: A/C/T, K: G/T, and M: A/C, where the underlined bold face indicates methylation sites) that were

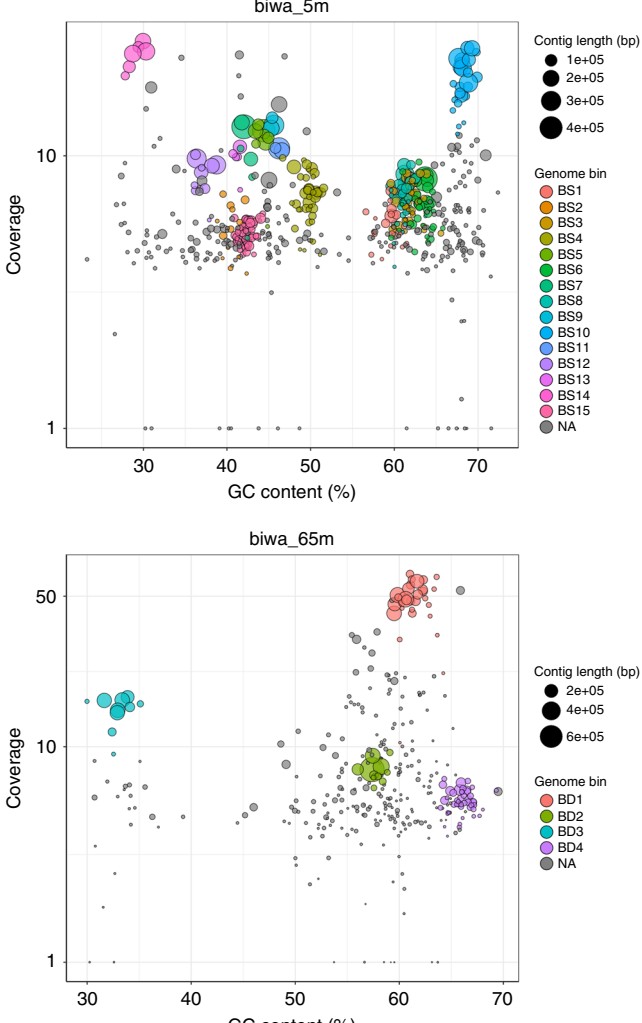

**Fig. 2** Genome binning of the assembled contigs. Each circle represents a contig, where the color and size represent its assigned bin and total sequence length, respectively. Contigs not assigned to any bin are indicated in gray (named 'NA'). The x-axis and y-axis represent GC% and genome coverage, respectively

likely due to incomplete detection of a single methylated motif or heterogeneous motif sequences between closely related lineages contained within that genome. A palindromic motif and five complementary motif pairs that likely reflect double-strand methylation were observed in the Bacteroidetes BS15 genome (e.g., a pair of **A**GCNNNNNNCAT and **A**TGNNNNNNGCT). It may also be notable that three draft genomes from the Chloroflexi phylum (BS1, BS3, and BD1) shared the same motif sequence set (G**A**NTC, TT**A**A, and GC**W**GC, where W: A/T), likely due to evolutionarily shared methylation systems. Contigs in each draft genome showed a similar methylation pattern in general, providing additional epigenomic support of the quality of the genome binning (Supplementary Fig. 6).

Overall, even if such similar, complementary, and shared motif sequences are considered, at least 9 motifs among the identified 22 motifs still presented no match to existing recognition sequences in the REBASE repository. This result demonstrates the existence of unexplored diversity of DNA methylation systems in environmental prokaryotes, which include many uncultured strains.

**Known MTases that correspond to detected methylated motifs.** To identify MTases that can catalyze the methylation reactions of the detected methylated motifs, systematic annotation of MTase genes was performed. Sequence similarity searches against known genes identified 20 MTase genes in nine draft genomes (sequence identities ranged from 23 to 71%) (Table 4). The most abundant group was Type II MTases, followed by Type I and Type III MTases, a trend that is consistent with the general MTase distribution[13,47]. Several genes encoding REases and DNA sequence-recognition proteins were also detected, and 9 of the 20 MTases (45%) were estimated to constitute RM systems (Table 4). The known motifs of 7 of the 20 MTases were matched to those identified in our metaepigenomic analysis (Table 3). For example, the Thaumarchaeota BD3 genome contained two MTases that showed the best sequence similarities to those that recognize AG**C**T and G**A**TC motif sequences, which were perfectly congruent with the two motifs detected in our metaepigenomic analysis. It may be notable that these two motifs were also reported in an enrichment-culture study of the closely related genus *Candidatus* Nitrosomarinus catalina[48] and are therefore likely evolutionarily conserved within their group. In the Proteobacteria BS14 genome, a similar one-to-one perfect match was also observed. The two genomes Chloroflexi BS3 and Chloroflexi BD1 were characterized by the same set of three methylated motifs, each of which contained three MTases. No MTase gene was found in the other Chloroflexi genome BS1, likely due to its low estimated genome completeness of 31% (Table 2). Among these MTases, two were most similar to those possessing methylation specificities that were congruent with two of the detected motifs, G**A**NTC and TT**A**A (the other MTase and motif will be discussed in the next section). Collectively, these observations suggest that metaepigenomic analysis is an effective tool for identifying the methylation systems of environmental prokaryotes.

**Unexplored diversity of prokaryotic methylation systems.** Among the 20 detected MTases, 13 MTases did not show sequence similarities to MTases that recognize the motifs identified in our metaepigenomic analysis (Tables 3 and 4). Although homology search-based MTase identification and recognition motif estimation are frequently conducted in genomic and metagenomic studies, this result suggests that these approaches are not sufficient, and direct observation of DNA methylation is needed to reveal the methylation systems of diverse environmental prokaryotes.

As noted earlier, each of the Chloroflexi BS3 and Chloroflexi BD1 genome had three MTase genes, two of which were congruent to two of the detected motifs. The other MTase from each genome (EMGBS3_12600 and EMGBD1_09320 in Chloroflexi BS3 and Chloroflexi BD1, respectively) showed the highest sequence similarity to an MTase that was reported to recognize A**C**GGC; however, the other methylated motif detected in the Chloroflexi BS3 and Chloroflexi BD1 genomes was GC**W**GC.

In the Bacteroidetes BS15 genome, 6 MTases and 11 methylated motifs were detected, but none of the MTases and motifs matched each other. At the methylation type level, five MTases and all of the methylated motifs were of the m6A type. We predicted that the EMGBS15_03820, whose closest homolog was an MTase that exhibits nonspecific m6A methylation activity, is actually a sequence-specific enzyme that recognizes a G**A**ANNNNTTC motif that was detected through metaepigenomic analysis, because the adjacent gene EMGBS15_03830 encodes an REase that targets the same GAANNNNTTC sequence.

### Table 2 Statistics for draft genomes

| Genome ID | Lineage | Estimated genome size (Mbp) | Contigs | N50 (bp) | GC content (%) | Completeness (%) | Contamination (%) | 16S rRNA | CDSs | CCS-read coverage | Methylated motifs | MTases |
|---|---|---|---|---|---|---|---|---|---|---|---|---|
| BS1 | Bacteria; Chloroflexi[a] | 2.24 | 21 | 64,528 | 59.5 | 30.6 | 0.0 | 0 | 751 | 5.79 | 3 | 0 |
| BS2 | Bacteria; Actinobacteria[a] | 1.57 | 13 | 28,617 | 40.6 | 16.9 | 0.0 | 0 | 363 | 5.13 | 0 | 0 |
| BS3 | Bacteria; Chloroflexi; Anaerolineae; Anaerolineales; Anaerolineaceae; uncultured; uncultured Crater Lake bacterium CL500-11 | 3.35 | 36 | 58,996 | 61.8 | 49.1 | 0.0 | 1 | 1646 | 6.91 | 3 | 3 |
| BS4 | Bacteria; Actinobacteria; Acidimicrobiia; Acidimicrobiales; Acidimicrobiaceae; CL500-29 marine group | 2.31 | 40 | 61,750 | 49.8 | 76.8 | 1.3 | 1 | 2066 | 6.67 | 0 | 0 |
| BS5 | Bacteria; Actinobacteria; Actinobacteria; Frankiales; Sporichthyaceae; hgcl clade; uncultured *Clavibacter* sp. | 1.51 | 8 | 190,417 | 44.2 | 71.6 | 0.0 | 1 | 1209 | 10.02 | 0 | 0 |
| BS6 | Bacteria; Verrucomicrobia; Opitutae; Opitutae vadinHA64; uncultured bacterium | 2.27 | 37 | 100,045 | 63.4 | 89.2 | 0.7 | 1 | 1889 | 6.85 | 0 | 1 |
| BS7 | Bacteria; Actinobacteria; Actinobacteria; Frankiales; Sporichthyaceae; hgcl clade; uncultured *Candidatus* Planktophila sp. | 1.49 | 6 | 470,028 | 42.1 | 58.4 | 0.6 | 1 | 948 | 9.26 | 0 | 0 |
| BS8 | Bacteria; Verrucomicrobia[b] | 2.71 | 34 | 102,020 | 61.2 | 82.5 | 2.0 | 0 | 2121 | 7.34 | 1 | 1 |
| BS9 | Bacteria; Actinobacteria[b] | 1.65 | 3 | 315,861 | 45.5 | 37.6 | 0.0 | 0 | 677 | 12.09 | 0 | 0 |
| BS10 | Bacteria; Verrucomicrobia; Opitutae; Opitutae vadinHA64; uncultured bacterium | 2.55 | 24 | 1,672,582 | 68.4 | 95.9 | 2.7 | 1 | 2165 | 17.93 | 1 | 1 |
| BS11 | Bacteria; Actinobacteria; Actinobacteria; Frankiales; Sporichthyaceae; hgcl clade; uncultured actinobacterium | 1.03 | 3 | 365,154 | 46.3 | 62.1 | 0.0 | 1 | 675 | 10.28 | 0 | 0 |
| BS12 | Bacteria; Proteobacteria; Betaproteobacteria; Methylophilales; Methylophilaceae; *Candidatus* Methylopumilus; uncultured bacterium | 1.40 | 10 | 169,468 | 37.3 | 80.7 | 0.4 | 1 | 1289 | 8.37 | 1 | 0 |
| BS13 | Bacteria; Actinobacteria; Actinobacteria[a] | 1.49 | 5 | 47,968 | 41.3 | 19.0 | 0.0 | 0 | 351 | 7.56 | 0 | 0 |
| BS14 | Proteobacteria; Alphaproteobacteria; Pelagibacterales[a] | 1.02 | 6 | 222,441 | 29.4 | 88.6 | 0.0 | 0 | 1075 | 20.45 | 1 | 1 |
| BS15 | Bacteria; Bacteroidetes; Sphingobacteriia; Sphingobacteriales; Chitinophagaceae; Filimonas; uncultured bacterium | 4.08 | 44 | 45,979 | 42.4 | 43.1 | 0.1 | 1 | 1908 | 5.57 | 6 | 6 |
| BD1 | Bacteria; Chloroflexi[a] | 2.89 | 30 | 157,947 | 60.9 | 90.9 | 0.9 | 0 | 2429 | 45.74 | 3 | 3 |
| BD2 | Bacteria; Nitrospirae[a] | 1.92 | 11 | 313,929 | 57.6 | 93.9 | 0.9 | 0 | 1890 | 8.01 | 1 | 2 |
| BD3 | Archaea; Thaumarchaeota; Marine Group I; Unknown Order; Unknown Family; *Candidatus* Nitrosoarchaeum | 1.48 | 10 | 250,506 | 33.0 | 98.5 | 1.9 | 1 | 1869 | 13.93 | 2 | 2 |
| BD4 | Bacteria; Verrucomicrobia[b] | 2.09 | 49 | 46,663 | 65.9 | 81.5 | 0.7 | 0 | 1705 | 5.98 | 0 | 0 |

[a]Estimated using CAT
[b]Estimated using Kaiju

In the Verrucomicrobia BS8 genome, one MTase and one methylated motif were detected; however, the reported recognition motif sequence of the closest MTase was incongruent with the detected motif (the reported and detected motifs were ACG**A**NNNNNNGRTC and **A**GGNNNNNRTTT, respectively, where R: A/G). This MTase is predicted to function in an RM system because of the existence of the neighboring REase and DNA sequence-recognition protein genes.

In the Verrucomicrobia BS10 genome, one MTase and one methylated motif were detected, and their motifs were also incongruent (GCA**A**GG and ACG**A**G, respectively).

In the Nitrospirae BD2 genome, two MTases and one methylated motif were detected. The two MTases EMGBD2_08760 and EMGBD2_08790 showed the best sequence similarities to those with m5C and m6A methylation activities, respectively, while the detected motif contained an m6A site.

Thus, the former MTase was predicted to catalyze the methylation reaction, although their motifs were again incongruent (GRGGA**A**G and TANGGA**B**, respectively). It should also be noted that these MTases appear to constitute a recently proposed system known as the Defense Island System Associated with Restriction-Modification (DISARM), which is a phage-infection defense system composed of MTase, helicase, phospholipase D, and DUF1998 genes[49]. To our knowledge, this is the first DISARM system identified in the phylum Nitrospirae.

In the Verrucomicrobia BS6 genome, one MTase gene was found, but we could not detect any methylated motif, and we therefore anticipate that this MTase gene does not exhibit methylation activity or the corresponding methylation motif was undetected due to the low sensitivity of SMRT sequencing to m5C modification as described previously[13,14]. However, in the Proteobacteria BS12 genome, we detected methylated motifs but

**Table 3 Detected methylated motifs**

| Genome ID | Detected methylated motif | Modification type | Motif in REBASE | Number of methylated sites | Number of motif sequences | Methylation ratio (%) | Mean modification QV | Mean subread coverage |
|---|---|---|---|---|---|---|---|---|
| BS1 | G**A**NTC | m6A | Yes | 1813 | 2070 | 87.6 | 58.0 | 35.2 |
| | TT**A**A | m6A | Yes | 1264 | 1522 | 83.0 | 55.5 | 34.1 |
| | GC**W**GC | m4C | Yes | 3026 | 15,948 | 19.0 | 38.4 | 40.6 |
| BS3 | G**A**NTC | m6A | Yes | 3724 | 4014 | 92.8 | 66.1 | 41.3 |
| | TT**A**A | m6A | Yes | 3036 | 3338 | 91.0 | 62.4 | 40.4 |
| | GC**W**GC | m4C | Yes | 13,821 | 54,026 | 25.6 | 39.5 | 46.4 |
| BS8 | **A**GGNNNNNRTTT | m6A | No | 80 | 276 | 29.0 | 39.6 | 65.8 |
| BS10 | ACG**A**G | m6A | No | 1986 | 7185 | 27.6 | 45.0 | 171.4 |
| BS12 | GMAG**C**TKC | m4C | No | 169 | 220 | 76.8 | 50.9 | 83.5 |
| | HCAG**C**TKC | m4C | No | 124 | 293 | 42.3 | 46.8 | 79.0 |
| | BGMAG**C**TGD | m4C | No | 78 | 185 | 42.2 | 46.3 | 76.3 |
| BS14 | G**A**NTC | m6A | Yes | 2856 | 2880 | 99.2 | 190.6 | 166.9 |
| BS15 | G**A**ANNNNTTC | m6A | Yes | 1309 | 1472 | 88.9 | 55.6 | 30.9 |
| | **A**GCNNNNNNCAT | m6A | No | 642 | 726 | 88.4 | 56.0 | 29.4 |
| | **A**TGNNNNNNGCT | m6A | No | 619 | 726 | 85.3 | 52.0 | 29.8 |
| | **A**GCNNNNNNGTG | m6A | No | 311 | 349 | 89.1 | 56.9 | 30.4 |
| | **C**ACNNNNNNGCT | m6A | No | 293 | 349 | 84.0 | 53.3 | 30.9 |
| | CA**A**NNNNNNNCTTG | m6A | No | 205 | 256 | 80.1 | 49.4 | 29.1 |
| | CA**A**GNNNNNNNDTTG | m6A | No | 164 | 214 | 76.6 | 48.7 | 28.7 |
| | TT**A**GNNNNNCCT | m6A | No | 87 | 99 | 87.9 | 51.3 | 29.8 |
| | **A**GGNNNNNCTAA | m6A | No | 77 | 99 | 77.8 | 49.4 | 29.7 |
| | GYT**A**NNNNNNNTTRG | m6A | No | 76 | 89 | 85.4 | 56.0 | 31.3 |
| | CYA**A**NNNNNNNTAVCH | m6A | No | 59 | 127 | 46.5 | 53.5 | 32.6 |
| BD1 | GC**W**GC | m4C | Yes | 72,730 | 77,932 | 93.3 | 140.2 | 297.3 |
| | G**A**NTC | m6A | Yes | 6754 | 6844 | 98.7 | 346.3 | 281.7 |
| | TT**A**A | m6A | Yes | 5475 | 5564 | 98.4 | 325.3 | 270.9 |
| BD2 | TANGG**A**B | m6A | No | 1276 | 1367 | 93.3 | 64.4 | 48.5 |
| BD3 | G**A**TC | m6A | Yes | 9446 | 9618 | 98.2 | 122.1 | 93.7 |
| | AG**C**T | m4C | Yes | 5974 | 6224 | 96.0 | 84.0 | 92.1 |

R = A/G, M = A/C, W = A/T, S = C/G, Y = C/T, K = G/T, H = A/C/T, B = C/G/T, D = A/G/T, V = A/C/G, N = A/C/G/T
Underlined bold face indicates methylation sites

no MTase genes. We assume that the MTase genes corresponding to this genome were missed due to insufficient genome completeness (although the estimated completeness was 81%), or because these MTase genes have diverged considerably from MTase genes found in cultivable strains, or because these MTases belong to a new group.

**Experimental verification of MTases with new methylated motifs**. Among the MTases whose sequences showed the best similarities to MTases that recognize motifs incongruent with our metaepigenomic results, we experimentally verified the methylation specificities of the four MTases: EMGBS3_12600 in Chloroflexi BS3 (and EMGBD1_09320 in Chloroflexi BD1, which has exactly the same amino-acid sequence), EMGBS15_03820 in Bacteroidetes BS15, EMGBS10_10070 in Verrucomicrobia BS10, and EMGBD2_08790 in Nitrospirae BD2 (Table 4). We constructed plasmids that each carried one of the artificially synthesized MTase genes, transformed them to *Escherichia coli* cells, forced their expression, and observed the methylation status of the isolated plasmid DNA by REase digestion.

Although the EMGBS3_12600 showed the best sequence similarity to a sequence-diverged MTase that possesses the A**C**GGC specificity, the unaccounted-for motif sequence observed in Chloroflexi BS3 was GC**W**GC. Thus, we hypothesized that the true recognition sequence of EMGBS3_12600 is GC**W**GC. The REase digestion assay showed that TseI (GCWGC specificity) did not cleave the plasmids when EMGBS3_12600 was expressed in the cells, which clearly supports our hypothesis (Fig. 3a). Furthermore, we confirmed that BceAI (ACGGC specificity) cleaved plasmids regardless of whether EMGBS3_12600 was

expressed, indicating that the EMGBS3_12600 protein does not show ACGGC sequence specificity (Fig. 3a). Accordingly, we named this protein M.AbaBS3I, as a novel MTase that possesses GC**W**GC specificity (Table 4).

While the homology-based analysis showed that the closest homolog of EMGBS15_03820 was a non-sequence-specific MTase, its adjacency to an REase and the results of the metaepigenomic analysis suggested that this MTase presents GA**A**NNNNTTC sequence specificity. The REase digestion assay showed that XmnI (GAANNNNTTC specificity) did not cleave the plasmids only when EMGBS15_03820 was expressed in the cells, which also supports our hypothesis (Fig. 3b). Furthermore, we confirmed that DpnII (GATC specificity) cleaved the plasmids regardless of whether EMGBS15_03820 was expressed, indicating that EMGBS15_03820 is not a nonspecific MTase. We named this protein M.FspBS15I, as a novel MTase that possesses GA**A**NNNNTTC methylation specificity (Table 4).

For EMGBS10_10070 in Verrucomicrobia BS10 and EMGBD2_08790 in Nitrospirae BD2, we also conducted REase digestion assays to confirm the recognition motif sequences. Based on the results of the metaepigenomic analysis, their motifs were predicted to be ACG**A**G and TANGG**A**B, respectively. Expression of each gene altered the electrophoresis patterns of the digested plasmids to contain fragments that resulted from inhibition of REase cleavage at the estimated methylation sites (Supplementary Fig. 7). Furthermore, we additionally conducted SMRT sequencing analysis using the PacBio RSII platform to examine the methylation status of the chromosomal DNA of the *E. coli* transformed with each of the two MTase genes. The results were basically consistent

**Table 4 Detected MTases, REases, and specificity subunit genes**

| Genome ID | CDS ID | Gene type | Top-hit protein in REBASE | Identity (%) | Recognition motif of the closest-match MTase | Modification type | RM type | RM system | TRD divergence | Motif detected | MTase name | Confirmed recognition motif |
|---|---|---|---|---|---|---|---|---|---|---|---|---|
| BS3 | EMGBS3_04270 | M | M.SstE37II | 58.9 | G**A**NTC | m6A | II | No | No | Yes | | |
| | EMGBS3_09240 | M | M. Sth20745I | 71.4 | TTA**A** | m6A | II | No | No | Yes | | |
| | EMGBS3_12600 | M | M1.BceSIII | 22.9 | A**C**GGC | m4C | II | No | Yes | No | M. AbaBS3I | G**C**WGC |
| BS6 | EMGBS6_08960 | M | M.SinI | 57.0 | GGW**C**C | m5C | II | No | No | No | | |
| BS8 | EMGBS8_10720 | R | DvuI | 36.3 | ? | – | I | – | – | – | | |
| | EMGBS8_10740 | S | S.PveNS15I | 32.4 | ? | – | I | – | Yes | – | | |
| | EMGBS8_10750 | M | M.RbaNRL2II | 55.6 | ACG**A**NNNNNNNGRTC | m6A | I | Yes | – | No | | |
| BS10 | EMGBS10_10070 | RM | CjeFIII | 23.7 | GCA**A**GG | m6A | II | Yes | Yes | No | M. ObaBS10I | ACG**A**G |
| BS14 | EMGBS14_10020 | M | M.Bsp460I | 56.7 | G**A**NTC | m6A | II | No | No | Yes | | |
| BS15 | EMGBS15_02830 | M | M.Bli37I | 56.6 | G**A**YNNNNNRTC | m6A | I | Yes | – | No | | |
| | EMGBS15_02840 | M | M.EcoNIH1III | 59.2 | G**A**TGNNNNNNTAC | m6A | I | Yes | – | No | | |
| | EMGBS15_02870 | S | S.PveNS15I | 47.2 | ? | – | I | – | Yes | – | | |
| | EMGBS15_02930 | R | DvuI | 38.4 | ? | – | I | – | – | – | | |
| | EMGBS15_03820 | M | M.EcoGI | 25.8 | Nonspecific | m6A | II | Yes | Yes | No | M. FspBS16I | G**A**ANNNNTTC |
| | EMGBS15_03830 | R | XmnI | 34.0 | GAANNNNTTC | – | II | – | – | – | | |
| | EMGBS15_04560 | R | GmeII | 33.8 | TCCAGG | – | III | – | – | – | | |
| | EMGBS15_04600 | M | M.FpsJII | 53.4 | CGC**A**G | m6A | III | Yes | No | No | | |
| | EMGBS15_05670 | M | M.FnuDI | 59.8 | GG**C**C[a] | m4C | II | Yes | No | No | | |
| | EMGBS15_05690 | R | BhaII | 45.6 | GGCC | – | II | – | – | – | | |
| | EMGBS15_12460 | M | M. Mva1261III | 37.1 | CT**A**NNNNNNRTTC | m6A | I | No | No | No | | |
| BD1 | EMGBD1_08400 | M | M. Sth20745I | 71.0 | TTA**A** | m6A | II | No | No | Yes | | |
| | EMGBD1_09320 | M | M1.BceSIII | 22.9 | A**C**GGC | m4C | II | No | Yes | No | M. AbaBS3I | G**C**WGC |
| | EMGBD1_19510 | M | M.SstE37II | 58.9 | G**A**NTC | m6A | II | No | No | Yes | | |
| BD2 | EMGBD2_08760 | M | M.HgiDII | 55.0 | GTCGAC[a] | m5C | II | Yes | No | No | | |
| | EMGBD2_08790 | RM | AquIV | 28.5 | GRGG**A**AG | m6A | II | Yes | Yes | No | M. NbaBD2I | TAHGG**A**B |
| | EMGBD2_08800 | R | LpnPI | 56.3 | CCDG | – | II | – | – | – | | |
| BD3 | EMGBD3_00670 | M | M. Mma5219II | 45.9 | AG**C**T | m4C | II | No | No | Yes | | |
| | EMGBD3_01960 | M | M.AvaVI | 50.3 | G**A**TC | m6A | II | No | No | Yes | | |

Underlined bold face indicates methylation sites
M: methyltransferase, R: restriction endonuclease, S: specificity subunit
[a]Modified base undetermined

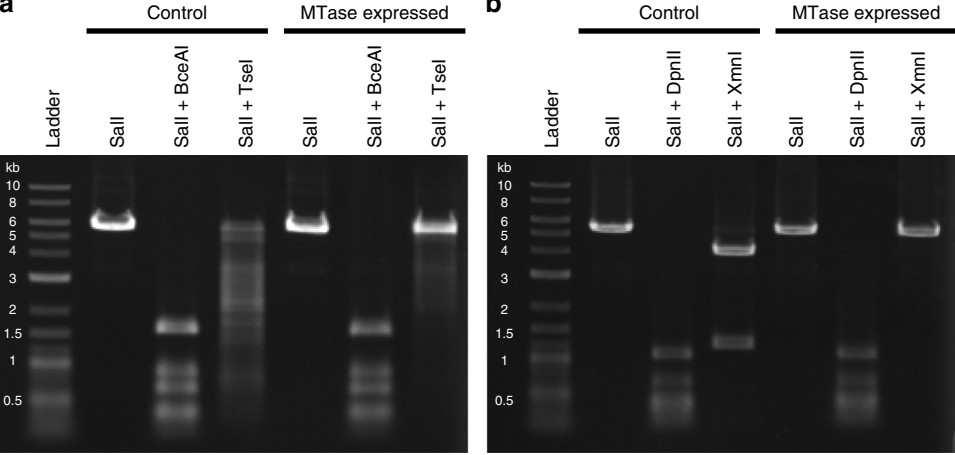

**Fig. 3** REase digestion assays. **a** Assay of the EMGBS3_12600 gene (and EMGBD1_09320, which has the same amino-acid sequence). BceAI and TseI were used, where the plasmid contained 12 (ACGGC) and 21 (GCWGC) target sites, respectively. Plasmid DNAs were linearized using SalI before the assay. An NEB 2-log DNA ladder was employed as a size marker. **b** Assay of the EMGBS15_03820 gene. DpnII and XmnI were used, where the plasmid contained 27 (GATC) and 2 (GAANNNNTTC) target sites, respectively

(Supplementary Table 4): ACG**A**G was actually detected as the methylated motif in *E. coli* transformed with EMGBS10_10070, and we named the protein M.ObaBS10I. In the case of EMGBD2_08790, the detected TAHGG**A**B motif was almost the same, but a subset of the estimated TANGG**A**B motif (i.e.,

TAGGG**A**B was excluded), and this difference could be due to *E. coli*-specific conditions (e.g., cofactors and sequence biases), insufficient data, inaccuracy of the methylated motif detection method. Regardless of this minor difference, we concluded that EMGBD2_08790 is a novel MTase gene responsible for

methylation of the TAHGG**A**B motif and we named the protein M.NbaBD2I accordingly.

**Metaepigenomics for exploring prokaryotic methylation systems in nature**. The present study demonstrated the effectiveness of the metaepigenomic approach powered by SMRT sequencing and CCS, showing obvious advantages over sequence similarity-based and culture-based methylation system analyses and short-read metagenomics. The CCS reads facilitated metagenomic assembly, binning, and protein sequence-based taxonomic assignment from an environmental sample that contained dominant uncultured prokaryotes. Most importantly, this approach revealed several methylated motifs, including novel ones in environmental prokaryotes, and subsequent experiments identified four MTases responsible for those reactions.

The current throughput of SMRT sequencing may be still insufficient to apply the metaepigenomic approach to more diverse and complex samples. Because deep sequencing coverage is required for the reliable detection of DNA methylation (for example, >25× subreads per each DNA strand is recommended according to the official instruction), it is still difficult to obtain sufficient sequencing reads to recover long contigs and detect methylated motifs for 'rare' species (typically those with <1% relative abundance). In addition to rapid and ongoing technological advances in SMRT sequencing, the emergence of Oxford Nanopore Technology may provide as another long-read, single-molecule, and methylation-detectable technology[50,51]. Another problem is that the detectable types of DNA modifications are limited (i.e., m4C, m5C, and m6A) with the currently available SMRT sequencing technology, while many other DNA chemical modifications occur in nature[52]. In addition to advances in sequencing methods, novel bioinformatic tools will be critical for metaepigenomic analyses of environmental prokaryotes.

A recent study showed that sets of methylated motifs and MTases can vary widely, even between closely related strains[53], where metaepigenomics is expected to enable differential methylation analyses between populations. It should be noted that metaepigenomic data may be adopted for various bioinformatic applications. For example, because reads and contigs in the same genome are expected to have the same methylation patterns, metaepigenomic information may be used for improving metagenomic assembly and binning[54]. In addition, genus-level conservation of MTases that are not associated with REases is sometimes observed, which suggests that MTases play unexplored adaptive roles, in addition to their functions in combating phages[13,55]. Novel MTases may be adopted for biotechnological uses, such as DNA recombination and methylation analyses[56]. It is envisioned that metaepigenomics of environmental prokaryotes under different sampling conditions and environments will significantly deepen our understanding of the ecological impacts of DNA methylation on prokaryotes, enigmatic evolution of prokaryotic methylation systems, and broaden their application potential.

## Methods

**Sample collection**. Water samples were collected at a pelagic long-term survey station (Ie-1) (35° 13′09.5″N 135°59′44.7″E) of the Center for Ecological Research, Kyoto University in Lake Biwa, Japan, on 26 December 2016 (Supplementary Fig. 1a). The sampling site was located approximately 3 km from the nearest shore and had a depth of 73 m. The lake has a permanently oxygenated hypolimnion and was thermally stratified during sampling (Supplementary Fig. 1b). Water sampling into prewashed 5-L Niskin bottles was conducted at depths of 5 m and 65 m, above and below the thermally stratified layer, respectively, to collect prokaryotic communities with different structures[34]. The vertical profiles of temperature, dissolved oxygen concentrations, and chlorophyll *a* concentrations were measured using a conductivity, temperature, and depth probe in situ. Equipment that could come into direct contact with the water samples in the following steps was either sterilized by autoclaving or disinfected with a hypochlorous acid solution. The water

samples were transferred to sterile bottles, kept cool by contact with ice packs in a dark cool box, and immediately transported to the laboratory. Water samples with a total volume of approximately 30 L were prefiltered through 5 μm membrane PC filters (Whatman). Microbial cells were collected using 0.22 μm Sterivex filters (Millipore) and immediately stored at −20 °C in a refrigerator until analysis.

**DNA extraction and SMRT sequencing**. The microbial DNA was retrieved using a PowerSoil DNA Isolation Kit (QIAGEN) according to the supplier's protocol with slight modifications as described below. The filters were removed from the container, cut into 3 mm fragments, and directly suspended in the extraction solution from the kit for cell lysis. The bead-beating time was extended to 20 min to yield sufficient quantities of DNA for SMRT sequencing, with reference to Albertsen et al.[57]. SMRT sequencing was conducted using a PacBio Sequel system (Pacific Biosciences) in two independent runs according to the manufacturer's standard protocols. SMRT libraries for CCS were prepared with a 4 kbp insertion length and two SMRT cells were used for each sample. Briefly, 3–5 kbp DNA fragments from each genomic DNA sample were extracted using the BluePippin size-selection system (Sage Science). Two sequencing libraries for CCS analysis were prepared using the SMRTbell Template Prep Kit 1.0-SPv3 according to the manufacturer's protocol (Pacific Biosciences). The final libraries were sequenced using a PacBio Sequel sequencer with Sequel SMRT Cell 1M v2 and Sequel Binding/Sequencing Kits 2.0.

**Bioinformatic analysis of CCS reads**. Reads that contained at least three full-pass subreads on each polymerase read were retained to generate CCS reads using the standard PacBio SMRT software package with the default settings. Only CCS reads with >97% average base-call accuracy were retained. For taxonomic assignment of the CCS reads, Kaiju[28] in *Greedy-5* mode with the NCBI nr database[29] and Kraken[30] with the default parameters and complete prokaryotic genomes from RefSeq[31] were used. CCS reads that potentially encoded 16S rRNA genes were extracted using SortMeRNA[58] with the default settings, and the 16S rRNA sequences were predicted by RNAmmer[59] with the default settings. The 16S rRNA sequences were taxonomically assigned using blastn[60] searches against the SILVA database release 128[61], where the top-hit sequences with e-values ≤ 1E−15 were retrieved.

CCS reads were de novo assembled using Canu[18] with the -pacbio-corrected setting and Mira[39] with the settings for PacBio CCS reads, according to the provided instructions. The Canu assembler provides information on repetitive contigs based on the graph topology and read-overlap analyses. Because such contigs are known to tend to contain misassembles, which can negatively affect accuracies of downstream analyses, we removed them. The remaining contigs were binned into genomes using MetaBAT[40] based on genome coverage and tetranucleotide frequencies as genomic signatures, where the genome coverage was calculated by mapping the CCS reads to the assembled contigs using BLASR[62] with the settings for PacBio CCS reads. The quality of all genomes was assessed using CheckM[63], which estimates completeness and contaminations based on taxonomic collocation of prokaryotic marker genes with the default settings. Sequence extraction and taxonomic assignment of 16S rRNA genes in each draft genome were conducted using RNAmmer[59] with the default settings. Taxonomic assignment of the draft genomes was based on the 16S rRNA genes if found or on the taxonomic groups most frequently estimated by CAT[64] otherwise (and Kaiju[28] if CAT did not provide an estimation).

Coding sequences (CDSs) in each draft genome were predicted using Prodigal[65] with the default settings. Functional annotations were achieved through GHOSTZ[66] searches against the eggNOG[67] and Swiss-Prot[68] databases, with a cut-off e-value ≤ 1E−5, and HMMER[69] searches against the Pfam database[70], with a cut-off e-value ≤ 1E−5. A maximum-likelihood tree of the draft genomes was constructed on the basis of the set of 400 conserved prokaryotic marker genes using PhyloPhlAn[71] with the default settings.

**Metaepigenomic and RM system analyses**. DNA modification detection and motif analysis were performed according to BaseMod (https://github.com/ben-lerch/BaseMod-3.0). Briefly, the subreads were mapped to the assembled contigs using BLASR[62], and interpulse duration ratios were calculated. Candidate motifs with scores higher than the default threshold value were retrieved as methylated motifs. Those with infrequent occurrences (<50) or very low methylation fractions (<1%) in each draft genome were excluded from further analysis. The methylated ratios of all detected motifs on each contig were calculated using Seqkit[72]. The sequence divergences of target recognition domains (TRDs) from those of the closest-match MTases were investigated using amino-acid alignments of BLASTP[60].

Genes encoding MTases, restriction endonucleases (REases), and DNA sequence-recognition proteins were detected by BLASTP[60] searches against an experimentally confirmed gold-standard dataset from the Restriction Enzyme Database (REBASE)[73] (downloaded on 2 October 2017), with a cut-off e-value of ≤ 1E−15. Sequence specificity information for each hit MTase gene was also retrieved from REBASE. The flanking regions of the MTase genes were investigated to search for REase genes and examine whether they constitute RM systems.

**Experimental verification of MTase activities**. For verification of the estimated methylation specificities, all four estimated Type II MTase genes (EMGBS3_12600, EMGBS15_03820, EMGBS10_10070, and EMGBD2_08790) that satisfied the following two criteria were selected: (1) their novel methylation motifs were uniquely predicted and (2) additional proteins were not required in evaluating their enzyme activities. The four MTases were artificially synthesized with codon optimization and cloned into the pUC57 cloning vector by Genewiz (Supplementary Data 1). The genes were subcloned into the pCold III expression vector (Takara Bio) using an In-FusionHD Cloning Kit (Takara Bio). The gene-specific oligonucleotide primers used for polymerase chain reaction and recombination are described in Supplementary Table 1. For verification of the EMGBS10_10070 gene function, the 5′-ACGAGTC-3′ sequence was inserted downstream of the termination codon for the sake of the methylation assay (the first five-base ACGAG sequence was the estimated methylated motif, and the last five-base GAGTC is recognized by the restriction enzyme PleI) (Supplementary Data 1).

The constructs were transformed into *E. coli* HST04 *dam⁻/dcm⁻* (Takara Bio), which lacks *dam* and *dcm* MTase genes. The *E. coli* strains were cultured in LB broth medium supplemented with ampicillin. MTase expression was induced according to the supplier's protocol. Plasmid DNAs were isolated using the FastGene Xpress Plasmid PLUS Kit (Nippon Genetics). SalI was employed to linearize the plasmid DNAs encoding EMGBS3_12600 and EMGBS15_03820 and then inactivated by heat. Methylation statuses were assayed by enzymatic digestion using the following restriction enzymes: BceAI and TseI for EMGBS3_12600, DpnII and XmnI for EMGBS15_03820, PleI for EMGBS10_10070, and FokI for EMGBD2_08790. All restriction enzymes were purchased from New England BioLabs. All digestion reactions were performed at 37 °C for 1 h, except for those involving TseI (8 h) and FokI (20 min). Notably, although TseI digestion is conducted at 65 °C in the manufacturer's protocol, we adopted a temperature of 37 °C to avoid cleavage of methylated DNA.

We further verified the methylated motifs that were newly estimated in this study, i.e., those of EMGBS10_10070 and EMGBD2_08790. Chromosomal DNA was extracted from cultures of the transformed *E. coli* strains using a PowerSoil DNA Isolation Kit (QIAGEN) according to the supplier's protocol. SMRT sequencing was conducted using PacBio RSII (Pacific Biosciences), and methylated motifs were detected via the same method described above.

## Data availability

The raw sequencing data and assembled genomes were deposited in the DDBJ Sequence Read Archive and DDBJ/ENA/GenBank, respectively (Supplementary Table 2). All data were registered under BioProject ID PRJDB6656.

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

## Acknowledgements

The sampling was conducted using Joint Usage/Research Grant of Center for Ecological Research, Kyoto University. The SMRT sequencing was supported by National Institute of Genetic, Research Organization of Information and Systems, Mishima, Japan. We thank Yoshinori Nii, Masashi Yoshino, and Satoko Fukuda for their helpful suggestions and experimental supports. We are grateful to Yukiko Goda and Tetsuji Akatsuka for their assistance in the field sampling, using Joint Usage / Research Grant of Center for Ecological Research, Kyoto University. We also thank Metabologenomics, Inc. for financial support. This work was supported by the Japan Science and Technology Agency (CREST), the Japan Society for the Promotion of Science (Grant Numbers 15J00971, 15J08604, 15H01725, 16H06154, and 17H05834), the Ministry of Education, Culture, Sports, Science, and Technology in Japan (221S0002 and 16H06279), and Leave a Nest Grant.

## Author contributions

S.H. conceived the study, performed the bioinformatics analyses and experiments, and wrote the manuscript. Y.O. and S.N. performed the water sampling. M.A. performed the experiments. A.T. performed the genomic and metagenomic sequencing. W.I. conceived the study, wrote the manuscript, and supervised the project. All authors read and approved the final manuscript.

## Additional information

**Competing interests:** The authors declare no competing interests.

