## [Peer Review File · Nature Communications]

Reviewers' comments:

Reviewer #1 (Remarks to the Author):

Review of Hiraoka et al.

Overall, this is a very good contribution, but some technical issues in the presentation need to be fixed. These mainly relate to the authors' predictions, which for the most part should not be phrased as predictions, but should merely be stated as the closest homologs of known recognition sequence. In almost every case the presentation makes it sound as though they have achieved more than they really have by disproving previous predictions, which is not the case. There was one disappointing aspect to the manuscript and that concerned the potential to use the metagenomic data to improve the accuracy of contig building. It would have been very nice to know if, within the bins obtained, the various long reads had all of the methylase motifs predicted from the methylase gene content and no methylated motifs from some or all of the other methylase genes present in the collection. i.e were the other recognition sequences present and unmodified? Tables 3 and 4 need to be redone in light of the comments below. In particular, in Table 3 the motifs such as AGCNNNNNCAT and AGCNNNNNGCT should be combined, since they are complementary strands of the same sequence and written in single-stranded form as the others are. This will affect the numbers of new motifs. In Table 4, the column "predicted recognition motif" should be relabeled as "closest match" and it would be useful to mark those where the target recognition domain is completely dissimilar. Also, the enzyme names should change in the MTase name column to reflect the fact that the strains as listed in the GenBank files differ from the ones that I suspect the authors used when they submitted them. Thus M.AspBS3I should be M.AbaBS3I and M.NspBD2I should be NbaBD2I. Unless of course the authors want to try and get the names changed back to their original designation, which would involve dealing with the GenBank (or DDBJ) staff.

I have some specific points detailed below.

1. 14 "novel motifs" (line 63. Also lines 243-244), but how many are genuine and this should be revised based on the comments about Table 4 above.
2. Line 278 (and 317). The ACGGC quote is not correct. The closest gold standard methylase is M2.BstI and recognizes GCAGTG. However, the level of similarity is 11% and there is no similarity in the target recognition domain. This just needs to be made clear.
3. Line 282 (and 325-326). The specificity of the 3820 methylase cannot be assigned at all by BLAST search as the closest gold standard is M1.BstF5I which has 12% similarity, but none at all in the target recognition domain. A closer examination of the BLAST results would have shown that there is a good hit to M.Fps101ORF3100P, which comes from an organism that also methylates GAANNNTTC .
4. Lines 286-290. I don't know where the ACGANNNNNGRTC came from, since neither the M gene nor the S gene has such a sequence marked as a close hit. Since the specificity is solely determined by the S subunit that is the only way that is relevant and it has no close hits with both TRDs, but does have one reasonable hit (S.Eox12280I – rec seq AAAYNNNNNGTG) where the TRD1 is identical to that observed in this bin.
5. Lines 291-292. Again, there is a problem with the statement about the closest hit, which is really only 30% similar and is Nal45188II – ACCAGC. No prediction can be made as there is no similarity whatsoever in the target recognition domains.
6. Lines 295-296. Again there is a problem with the statement about the closest hit, which is AcoY31II (TAGCRAB), but there are many differences in the target recognition sequences in the two Type IIG genes.
7. Line 319-324. TseI has never been explicitly tested against GCWGC where the modification is m4C, but it can be inferred from the fact that the native M.TseI is an m4C methylase. The BceAI experiment really says nothing useful as the ACGGC specificity suggested by the authors was a baseless speculation.
8. Lines 325-332. The XmnI experiment is a good one, but the text about this being a non-specific

methylase is wrong and the DpnII experiment is irrelevant.

9. Lines 341-346. It seems likely based on my experience that the difference between the genome prediction and the clone is the degree of coverage. The genome result is more believable from two perspectives. One is that the % methylation is very high and the coverage of 48.5 % is much more likely to give reliable answers. In the case of the clone the higher coverage (138 fold) and the lower % of methylation suggests a software problem. However, the authors' explanation that it may be an expression problem cannot be excluded.

Rich Roberts

Reviewer #2 (Remarks to the Author):

In this manuscript, Hiraoka and colleagues analyze the methylation landscape of a lake metagenome. The aim of this research is to: a. characterize the taxonomic composition of the microbiome in the lake. b. identify methylation sites in the bacterial genomes that were assembled successfully. c. link the methyltransferase genes in assembled genomes to observed methylated sites.

To do this the authors did the following: First, they sequenced lake samples in two locations using the SMRT PacBio sequencing platform. They then taxonomically classified the reads up to class level and assembled the reads into contigs and obtained draft genomes. They used the PacBio sequencing platform to detect methylation motifs in each genome and then based on sequence homology to known Restriction-Modification (RM) systems, identified R, M and Specificity genes in each genome. They assigned methylation motif to methyl transferase genes according to predicted methylation site as presented in the REBASE database. Finally, for a subset of methyltransferase genes for which the observed methylation motifs was not consistent with REBASE prediction of the motif, they experimentally show that the motif detected by PacBio is correct whereas the REBASE prediction is incorrect.

They report that the taxonomic composition of the lake is consistent with other studies of fresh water environments, with Proteobacteria being the most prevalent phyla detected. Nineteen draft genomes originating from 7 phyla were assembled, on which they perform the methylation analysis. 29 methylated motifs were detected among 10 of the 19 draft genomes. 14 of the 29 motifs are not represented in REBASE. Using similarity search against a database of RM genes, they identify 20 methyl transferase genes in 9/19 of the draft genomes. They compare the predicted motifs of these 20 genes to the observed methylated sites and see that only 7/20 motifs matched the motifs they could identify using PacBio. For 4 of the 13 methylases for which the recognition motif did not match the Rebase prediction, they experimentally show that the motif identified in their study is correct.

Overall, this paper describes a scientifically sound investigation of an interesting and relatively novel scientific area, that of metaepigenomics. While many bacteria have been sequenced with the PacBio platform as individual isolates (Blow et al, PLoS Genetics, 2016), little has been done to evaluate the epigenetic landscape of uncultured bacteria in complex microbiome mixtures. The work presented here identifies novel methyltransferases, as outlined above, and presents convincing experimental results supporting the predicted enzymatic activity of these gene products.

I have several suggestions, both major and minor, that if addressed would increase the clarity and impact of this manuscript.

Major:

1. The authors state that coverage of at least 25 is needed for accurate methylation calling (line 383), but only one genome bin has >25X coverage as per table 2. Also, the motif coverages that are reported in Table 3 are much, much higher than the coverage in the genome bins. How is this possible? This seems contradictory to me.
2. Most reads were mapped to Proteobacteria, but in the draft genomes we see only two draft genomes that originate from Proteobacteria. Does that mean that most of the Proteobacteria reads originate from only two apparently very abundant Proteobacteria species?
3. Once draft genomes are assembled - what is the percentage of reads that now map back to these genomes? In other words, how well do these 19 draft genomes represent what is in the lake?
4. The authors evaluate phage infection through prophage detection. This is a very limited evaluation since phages are known to appear in two forms - temperate or virulent phages. Virulent phages do not integrate into the genome and thus cannot be detected via prophage search. In addition, recent work (Doron et al, Science, 2018) suggest that in addition to the well-known RM and CRISPR systems, multiple other anti-phage systems are encoded in bacterial genomes. Therefore, since RM systems are only one of many potential anti-phage systems and since the ability to measure phage infection at this setting is very limited, it cannot be concluded that lack of RM systems is linked in this small set of genomes to lack of phage pressure. Furthermore, the prevalence of prophage in the bins without substantial methylation (line 355) is not evaluated. This would seem to be an important control in this analysis.
5. Line 105 – why were contigs with repeats excluded from binning and how were these repeat-containing contigs identified? Most genomes contain repetitive elements (for example, 16S rRNA sequence – often present in multiple copies), thus, excluding contigs with any repeats seems inappropriate.

Minor:

6. Overall, the paper would benefit from spelling and grammar overview. For example: line 42: "...most studies must focus..."
7. Samples were collected from two sites – if would be nice for the general reader if the authors clarify the biological reason for studying the two sites and the implications/expectations of doing so.
8. Can the authors conclude anything about sequence similarity required between two MTase genes to maintain the methylated motif? 23% similarity is low, it may be expected to have a different motif.
9. Experimental verification of MTases with new methylated motifs – why did the authors choose these 4 MTases to characterize? Is it expected that for the remaining 9 the methylation motif is also consistent with PacBio results or is there some unique attribute to the 4 MTase chosen that make it more probable for them to recapitulate the PacBio results and not the Rebase predictions?
10. According to the assembled contigs - which of the methyltransferase genes is part of a RM system and which is 'orphan'?
11. Table 4 would be more informative if added a column showing whether PacBio methylation observed is in line with prediction.
12. Line 235: HCAGCTKC is actually not overlapping with GMAGCTKC, since H cannot be G.
13. Semantically, 'draft genome' could be used instead of 'genome bin'. In addition, instead of random letters and numbers, throughout the manuscript it is better to keep the name of the genome at least in the Phylum level (e.g. Proteobacteria1).
14. Unclear sentences, like line 61 " Freshwater habitats are rich in phage-prokaryotic interactions, which are known to be closely related to prokaryotic DNA methylation...".
15. It is not accurate that the E.coli HST04 strain lacks endogenous MTase. It is only deleted for the dam and dcm genes, but still retains two other MTases (<http://rebase.neb.com/cgi-bin/acroegct?EcoHST04+o24091>)
16. Line 85 – "...slight modifications" – what are the modifications?
17. Line 169: should be "reads" and line 179 should be "subreads".

18. Methods – how was size selection performed? Also, how were SMRT sequencing libraries created. More experimental detail would be required for a reader to know how to reproduce these results.

19. Line 84 – Microbes were collected on the Sterivex filters, not DNA.

20. Lines 89-90 – how were CCS libraries prepared? Most readers will not be familiar with this method, so additional details as well as the purpose of this type of library creation will be important for the general audience.

21. I would not consider running the same library on 2 SMRT cells to really be a “technical replicate”.

22. Figure 1 – reads should be corrected for read size in this figure. If this has already been done, please clarify in the figure legend.

23. Figure 2 – what is the size cutoff for inclusion in this figure?

24. Figure S7 – There is inadequate detail in the figure legend or manuscript for me to interpret this figure.

#####

Reviewer #1:

#####

//=====

Overall, this is a very good contribution, but some technical issues in the presentation need to be fixed.

//=====

We would like to express our appreciation to the reviewer for the insightful comments on our manuscript.

//=====

These mainly relate to the authors' predictions, which for the most part should not be phrased as predictions, but should merely be stated as the closest homologs of known recognition sequence. In almost every case the presentation makes it sound as though they have achieved more than they really have by disproving previous predictions, which is not the case.

//=====

As per the reviewer's request, we have fixed all sentences regarding the "predictions" as follows:

L276: For example, the **Thaumarchaeota** BD3 **genome** contained two MTases **that showed the best sequence similarities to those that recognize AGCT and GATC motif sequences**, which were perfectly congruent with the two motifs detected in our metaepigenomic analysis.

L284: Among these MTases, two **were most similar to those possessing** methylation specificities that were congruent with two of the detected motifs, **GANTC** and **TTAA** (the other MTase and motif will be discussed in the next section).

L291: Among the 20 detected MTases, 13 MTases did not **show sequence similarities to those recognizing** motifs that matched those identified in our metaepigenomic analysis (Tables 3 and 4)

L303: We predicted that EMGBS15_03820, **whose closest homolog was an MTase that exhibits** non-specific m6A methylation activity, is actually a sequence-specific enzyme that recognizes a **GAANNNTTC** motif that was detected through metaepigenomic analysis, because the adjacent gene EMGBS15_03830 encodes an REase that targets the same GAANNNTTC sequence.

L308: however, the **reported recognition motif sequence of the closest** MTase was incongruent with the detected motif (the **reported** and detected motifs were **ACGANNNNNGRTC** and **AGGNNNNNRRTT**, respectively, where R: G/A).

L315: The two MTases **showed the best sequence similarities to those with** m6A and m5C

methylation activities, while the detected motif contained an m6A site.

L332: Among the MTases whose sequence showed best similarity to ones that recognize motif that were incongruent with our metaepigenomic results, we experimentally verified the methylation specificities of the four MTases

L339: Although EMGBS3_12600 showed the best sequence similarity to an MTase that possesses ACGGC specificity, their TRDs were divergent and the unaccounted-for motif sequence observed in Chloroflexi BS3 was GCWGC.

L348: While the homology-based analysis showed that the closest homolog of EMGBS15_03820 was a non-sequence specific MTase, its adjacency to an REase and the results of the metaepigenomic analysis suggested that this MTase presents GAANNNTTC sequence specificity.

//=====

There was one disappointing aspect to the manuscript and that concerned the potential to use the metagenomic data to improve the accuracy of contig building. It would have been very nice to know if, within the bins obtained, the various long reads had all of the methylase motifs predicted from the methylase gene content and no methylated motifs from some or all of the other methylase genes present in the collection. i.e were the other recognition sequences present and unmodified?

//=====

As per the reviewer's request, we visualized the ratios of all methylated motifs detected in this study on every contig (new Fig. S6). Contigs in each genome bin showed a similar methylation pattern in general, providing additional "epigenomic support" of the quality of the genome binning; however, several contigs showed methylation patterns that seem to be inconsistent to those of the other contigs in the same genome bin. As thoughtfully pointed out by the reviewer, these inconsistencies can be due to errors in the genome binning and be used for improving the assembly and binning. We, therefore, further examined such contigs in depth, but could not find any case that the inconsistency was concluded to be due to a binning error instead of a methylation detection error. For instance, the ~10 kbp contig in the BS15 draft genome where methylated ACGAG motif was detected (new Fig. S6) contained only two ACGAG sequences on the contig; that is, two methylation detection errors could also explain the observed inconsistency. In addition, the global detection of low-level methylation of the GMAGCTKC and HCAGCTKC motifs in the BS1 contigs and AGCT in the BD1 contigs were likely due to methylation of the overlapping GCWGC motif. According to these observations, we are keeping the binning results in the original manuscript at this revision. Instead, we have added new Fig. S6 (with revised Methods and Results) and a description about the potential use of the methylation information in improving the assembly and binning as follows. We would like to thank the reviewer again for pointing out this important issue.

L133: The methylated ratios of all detected motifs on each contig were calculated using Seqkit⁴⁹

L261: Contigs in each draft genome showed a similar methylation pattern in general, providing additional epigenomic support of the quality of the genome binning (Fig. S6).

L394: It should be noted that metaepigenomic data may be adopted for various bioinformatic applications. For example, because reads and contigs in the same genome are expected to have the same methylation patterns, metaepigenomic information may be used for improving metagenomic assembly and binning⁷¹.

Figure S6. Ratios of methylated motifs on each contig. The green color represents the methylation ratios of all motifs detected in this study. If a motif sequence is absent from any contig, the ratio was regarded as zero. While contigs in each draft genome showed a similar methylation pattern in general, several contigs showed methylation patterns that

seem to be inconsistent to those of the other contigs in the same genome bin. These inconsistencies could be due to errors in the genome binning and be used for improving the assembly and binning; however, there was no case that the inconsistency was concluded to be due to a binning error instead of a modification detection error. For instance, the ~10 kbp contig of the BS15 genome on which methylated ACGAG was detected contained only two ACGAG sequences, that is, two modification detection errors could also explain the observed inconsistency. In addition, the global detection of low-level methylation of GMAGCTKC and HCAGCTKC motifs in the BS1 contigs and AGCT in the BD1 contigs were likely due to methylation of the overlapping GCWGC motif.

//=====

In particular, in Table 3 the motifs such as AGCNNNNNNNCAT and AGCNNNNNNNGCT should be combined, since they are complementary strands of the same sequence and written in single-stranded form as the others are. This will affect the numbers of new motifs.

1. 14 “novel motifs” (line 63. Also lines 243-244), but how many are genuine and this should be revised based on the comments about Table 4 above.

//=====

As suggested, we have revised Table 3 to specify potentially complementary motifs and updated the numbers of novel motifs mentioned in the manuscript and Table 2. The total numbers of the identified methylated motifs and novel motifs have been changed to be 22 and 9, respectively.

L26: The analysis of DNA chemical modifications identified 22 methylated motifs in those genomes, among which 9 motifs were novel.

L63: CCS analyses of the environmental microbial samples allowed reconstruction of draft genomes and the identification of their methylated motifs, at least 9 of which were novel.

L263: Overall, even if such similar, complementary, and shared motif sequences are considered, at least 9 motifs among the identified 22 motifs still presented no match to existing recognition sequences in the REBASE repository.

//=====

In Table 4, the column “predicted recognition motif” should be relabeled as “closest match” and it would be useful to mark those where the target recognition domain is completely dissimilar.

//=====

As per the reviewer’s request, we have revised Table 4 by changing the label and examining alignments between TRDs of MTases identified in this study and the closest-match reference MTases

(and those between the corresponding S proteins if the MTases were Type I).

“Predicted recognition motif” -> “Recognition motif of the closest-match MTase”
(New column) “TRD divergence”

//=====

the enzyme names should change in the MTase name column to reflect the fact that the strains as listed in the GenBank files differ from the ones that I suspect the authors used when they submitted them. Thus M.AspBS3I should be M.AbaBS3I and M.NspBD2I should be NbaBD2I. Unless of course the authors want to try and get the names changed back to their original designation, which would involve dealing with the GenBank (or DDBJ) staff.

//=====

We express our appreciation to this careful comment; we have accordingly renamed the MTases as follows:

M.AspBS3I -> M.AbaBS3I
M.OspBS10I -> M.ObaBS10I
M.NspBD2I -> M.NbaBD2I

Because BS10 genome was registered as a *Filimonas* sp. strain, we are keeping the name of the EMGBS15_03820 MTase as M.FspBS16I.

//=====

2. Line 278 (and 317). The ACGGC quote is not correct. The closest gold standard methylase is M2.BstI and recognizes GCAGTG. However, the level of similarity is 11% and there is no similarity in the target recognition domain. This just needs to be made clear.
3. Line 282 (and 325-326). The specificity of the 3820 methylase cannot be assigned at all by BLAST search as the closest gold standard is M1.BstF5I which has 12% similarity, but none at all in the target recognition domain. A closer examination of the BLAST results would have shown that there is a good hit to M.Fps101ORF3100P, which comes from an organism that also methylates GAANNNTTC .
4. Lines 286-290. I don't know where the ACGANNNNNGRTC came from, since neither the M gene nor the S gene has such a sequence marked as a close hit. Since the specificity is solely determined by the S subunit that is the only way that is relevant and it has no close hits with both TRDs, but does have one reasonable hit (S.Eox12280I – rec seq AAAYNNNNNGTG) where the TRD1 is identical to that observed in this bin.
5. Lines 291-292. Again, there is a problem with the statement about the closest hit, which is really only 30% similar and is Nal45188II – ACCAGC. No prediction can be made as there is no similarity whatsoever in the target recognition domains.

6. Lines 295-296. Again there is a problem with the statement about the closest hit, which is AcoY31II (TAGCRAB), but there are many differences in the target recognition sequences in the two Type IIG genes.

7. Line 319-324. TseI has never been explicitly tested against GCWGC where the modification is m4C, but it can be inferred from the fact that the native M.TseI is an m4C methylase. The BceAI experiment really says nothing useful as the ACGGC specificity suggested by the authors was a baseless speculation.

8. Lines 325-332. The XmnI experiment is a good one, but the text about this being a non-specific methylase is wrong and the DpnII experiment is irrelevant.

//=====

According to these comments, we re-examined our dataset using the latest REBASE Gold Standards database (downloaded on Sep. 19, 2018 from <http://rebase.neb.com/cgi-bin/seqsget?L+qd>); however, we could not obtain the results described in this comment. Besides, we could not find one of the MTase (M2.BstI) from the REBASE portal site (<http://rebase.neb.com/rebase/rebase.html>) using the “search enzyme names” option and three MTases (S.Eox12280I, Nal45188II, and AcoY31II) from the All REBASE Gold Standards database (<http://rebase.neb.com/cgi-bin/seqsget?L+qd>). We are suspecting if there was any technical difference between the reviewer’s and our analyses (*e.g.*, database version or BLAST options). We will be willing to update the analyses if the details regarding this issue become clear, while we would like to note that it will not affect the main story of this manuscript. At this revision, to clarify our method of the methylated motif identification, we have added the date when we downloaded the REBASE database.

L137: Genes encoding MTases, restriction endonucleases (REases), and DNA sequence-recognition proteins were detected by BLASTP³⁴ searches against an experimentally confirmed gold-standard dataset from the Restriction Enzyme Database (REBASE)⁴⁸ (downloaded on Oct 2, 2017), with a cut-off e-value of $\leq 1E-15$.

//=====

9. Lines 341-346. It seems likely based on my experience that the difference between the genome prediction and the clone is the degree of coverage. The genome result is more believable from two perspectives. One is that the % methylation is very high and the coverage of 48.5 % is much more likely to give reliable answers. In the case of the clone the higher coverage (138 fold) and the lower % of methylation suggests a software problem. However, the authors’ explanation that it may be an expression problem cannot be excluded.

//=====

We thank the reviewer very much for providing us with this insight. Because the true cause cannot be identified as suggested, we would like to keep the original discussion.

#####

Reviewer #2 (Remarks to the Author):

#####

//=====

Overall, this paper describes a scientifically sound investigation of an interesting and relatively novel scientific area, that of metaepigenomics. While many bacteria have been sequenced with the PacBio platform as individual isolates (Blow et al, PLoS Genetics,2016), little has been done to evaluate the epigenetic landscape of uncultured bacteria in complex microbiome mixtures. The work presented here identifies novel methyltransferases, as outlined above, and presents convincing experimental results supporting the predicted enzymatic activity of these gene products.

//=====

We really appreciate this encouraging comment and thank the reviewer for the insightful comments so much.

//=====

1. The authors state that coverage of at least 25 is needed for accurate methylation calling (line 383), but only one genome bin has >25X coverage as per table 2. Also, the motif coverages that are reported in Table 3 are much, much higher than the coverage in the genome bins. How is this possible? This seems contradictory to me.

//=====

We are sorry for the confusing descriptions. First, coverages in Tables 2 and 3 are calculated based on CCS reads and subreads, respectively (the latter is much larger). Second, the >25X coverage per strand is the optimal subread coverage that is officially required by the manufacturer (Ref: <https://github.com/PacificBiosciences/Bioinformatics-Training/wiki/Methylome-Analysis-Technical-Note>). To clarify these points, we have revised the manuscript as below:

Table 2 column label: **CCS-read** coverage

Table 3 and Table S5 column label: **Mean subread coverage**

L252: **The mapped subread coverages of the methylated motifs ranged from 28.7 to 297.3×.**

L383: Because deep sequencing coverage is required for the reliable detection of DNA methylation **(for example, >25× subreads per each DNA strand is recommended according to the official instruction)**, it is still difficult to obtain sufficient sequencing reads to recover long contigs and detect methylated motifs for 'rare' species (typically those with

<1% relative abundance).

//=====

2. Most reads were mapped to Proteobacteria, but in the draft genomes we see only two draft genomes that originate from Proteobacteria. Does that mean that most of the Proteobacteria reads originate from only two apparently very abundant Proteobacteria species?

//=====

The two Proteobacteria draft genomes (BS12 and BS14) likely belong to clades known as “Candidatus Methylopumilus (LD28)” and “LD12”, respectively, which were reported to be the most abundant Proteobacteria clades in the Biwa lake (Okazaki et al. Env Microbiol Rep, 2016; Okazaki et al., ISME J, 2017). The numbers of CCS reads that could be mapped to BS12 and BS14 genome bins were 5,107 and 7,124, respectively, and the number of CCS reads that were assigned to Proteobacteria were 46,914. Thus, approximately 25% of the CCS reads from the shallow sample are estimated to have originated from the BS12 and BS14 genome bins. We have added these explanations to the Result section as below:

L242: Although Proteobacteria is the most dominated phyla, two and none draft genomes were retrieved from the shallow and deep samples. Regarding the shallow sample, approximately one-fourth of the Proteobacteria CCS reads could be mapped to the two draft genomes, that means, three-fourths of them likely originated from minor and diverse Proteobacteria clades.

//=====

3. Once draft genomes are assembled - what is the percentage of reads that now map back to these genomes? In other words, how well do these 19 draft genomes represent what is in the lake?

//=====

As per the reviewer’s request, we have calculated the ratios of the CCS reads mapped to the BS1-15 (for biwa_5m) and BD1-4 (for biwa_65m) genome bins, and they were 46.9 and 44.8%, respectively. We have added this result to the Result section as follows:

L224: In total, 46.9 and 44.8% of the CCS reads could be mapped to the draft genomes for the shallow and deep samples, respectively.

//=====

4. The authors evaluate phage infection through prophage detection. This is a very limited evaluation since phages are known to appear in two forms - temperate or virulent phages. Virulent phages do not integrate into the genome and thus cannot be detected via prophage search. In addition, recent work (Doron et al, Science, 2018) suggest that in addition to the well-known RM and CRISPR

systems, multiple other anti-phage systems are encoded in bacterial genomes. Therefore, since RM systems are only one of many potential anti-phage systems and since the ability to measure phage infection at this setting is very limited, it cannot be concluded that lack of RM systems is linked in this small set of genomes to lack of phage pressure. Furthermore, the prevalence of prophage in the bins without substantial methylation (line 355) is not evaluated. This would seem to be an important control in this analysis.

//=====

We thank the reviewer very much for providing these insights. We understood and agreed with the reviewer that our discussion on the prophage analyses require many assumptions; therefore, we decided to remove the prophage analysis parts from the Abstract, Method, Results, and Conclusion sections from our manuscript (with Figure S7 and Table S6 from supplemental information). We would like to note that these removals do not affect the main story of this study.

//=====

5. Line 105 – why were contigs with repeats excluded from binning and how were these repeat-containing contigs identified? Most genomes contain repetitive elements (for example, 16S rRNA sequence – often present in multiple copies), thus, excluding contigs with any repeats seems inappropriate.

//=====

Excluding or including contigs with repeats can actually have pros and cons. As pointed out by the reviewer, contigs with repeats can contain, for example, 16S rRNA genes, which are useful for many purposes. On the other hand, such contigs tend to contain misassemblies, which can negatively affect accuracies of downstream bioinformatic analyses (*e.g.*, the genome binning and methylated motif identification processes, which were the keys in this study) (Olson et al. Briefings in Bioinformatics, 2017). Because our aim here was to provide reliable metaepigenomic data as much as possible, we took the latter, conservative approach. Namely, as the Canu assembler provides information on repetitive contigs based on the graph topology and read-overlap analyses, we ignored those contigs. Besides, we deposited raw data in a public database for those who are interested in analyzing such repetitive sequences. We have inserted the following information about the repeated contig exclusion in the Methods section.

L110: The Canu assembler provides information on repetitive contigs based on the graph topology and read-overlap analyses. Because such contigs are known to tend to contain misassemblies, which can negatively affect accuracies of downstream analyses, we removed them. The remaining contigs were binned into genomes using ~.

//=====

6. Overall, the paper would benefit from spelling and grammar overview. For example: line 42: "...most studies must focus..."

//=====

Although our manuscript had been edited by Springer Nature English Editing Service before the submission, we apologize that some mistakes seem to have remained. We have revised the sentence as below:

L41: deleted "must"

//=====

7. Samples were collected from two sites – if would be nice for the general reader if the authors clarify the biological reason for studying the two sites and the implications/expectations of doing so.

//=====

First of all, freshwater lakes are of economical and social importance, where microbes constitute the bases of those ecosystems (Newton et al. *Microbiol. Mol. Biol. Rev.* 2011). Second, the Lake Biwa sampling site has been a long-term sampling station (Ie-1) since 1965; thus, Illumina-based microbial-community and other (e.g., chemical) data were available for comparison, as we did in the original manuscript. Third, the community was known to contain many microbial groups that have not been cultured but be not too complicated, and these characteristics were ideal for the proof-of-concept study of metaepigenomics. To clarify these, we have revised the Introduction, Method, and the Conclusion sections.

L59: Freshwater lakes are of economical and social importance, where microbes constitute the bases of their ecosystems²³.

L70: Water samples were collected at a pelagic long-term survey station (Ie-1) (35° 13' 09.5" N 135° 59' 44.7" E) of the Center for Ecological Research, Kyoto University in Lake Biwa, Japan on December 26, 2016 (Fig. S1a).

L74: Water sampling into prewashed 5-L Niskin bottles was conducted at depths of 5 m and 65 m, above and below the thermally stratified layer, respectively, to collect prokaryotic communities with different structures²⁷.

L400: It is envisioned that metaepigenomics of environmental prokaryotes under different sampling conditions and environments will significantly deepen our understanding of the ecological impacts of DNA methylation on prokaryotes, enigmatic evolution of prokaryotic methylation systems, and broaden their application potential.

//=====

8. Can the authors conclude anything about sequence similarity required between two MTase genes to maintain the methylated motif? 23% similarity is low, it may be expected to have a different

motif.

//=====

The short answer to this question is no. The similarity of the target recognition domain (TRD) of MTases can give a clue to predict motif specificities of MTases, but it is also reported that a few amino-acid substitutions in a TRD can lead to changes in its motif specificity (Szegedi and Gumpert, Nucleic Acids Res, 2000). Thus, at this revision, we have revised Table 4 by constructing alignments between TRDs of MTases identified in this study and the closest-match reference MTases (and those between the corresponding S proteins if the MTases were Type I). Column “Sequence divergence of recognition domain from closest-match MTase” has been added in Table 4 to indicate whether large sequence divergences were observed or not. We have revised the Methods and Results sections as follows:

L134: The sequence divergences of target recognition domains (TRDs) from those of the closest-match MTases were investigated using amino-acid alignments of BLASTP³⁵

L339: Although EMGBS3_12600 showed the best sequence similarity to a sequence-diverged MTase that possesses the ACGGC specificity, the unaccounted-for motif sequence observed in Chloroflexi BS3 was GCWGC.

//=====

9. Experimental verification of MTases with new methylated motifs – why did the authors choose these 4 MTases to characterize? Is it expected that for the remaining 9 the methylation motif is also consistent with PacBio results or is there some unique attribute to the 4 MTase chosen that make it more probable for them to recapitulate the PacBio results and not the Rebase predictions?

//=====

In this study, we chose all four MTases that satisfied the following two criteria: (1) Their novel methylation motifs were uniquely predicted and (2) additional proteins were not required in evaluating their enzyme activities (e.g., a Type I MTase requires a corresponding S subunit protein for DNA methylation, where problems in transformation, concerted protein expression, or complex formation could inhibit the methylation assay experiments (Srikhanta et al. 2010)). We have accordingly revised the Method section as follows:

L145: For verification of the estimated methylation specificities, all four estimated Type II MTase genes (EMGBS3_12600, EMGBS15_03820, EMGBS10_10070, and EMGBD2_08790) that satisfied the following two criteria were selected: (1) Their novel methylation motifs were uniquely predicted and (2) additional proteins were not required in evaluating their enzyme activities. The four MTases were artificially synthesized with codon optimization and cloned into the pUC57 cloning vector by Genewiz (Table S1).

//=====

10. According to the assembled contigs - which of the methyltransferase genes is part of a RM system and which is 'orphan'?

//=====

As the reviewer's request, we have added a new column "RM system" to Table 4 (if "yes", corresponding REase gene was found). In addition, we have accordingly revised the Method and Results sections as follows:

L140: The flanking regions of the MTase genes were investigated to search for REase genes and examine whether they constitute RM systems.

L273: Several genes encoding REases and DNA sequence-recognition proteins were also detected, and nine of the 20 MTases (45%) were estimated to constitute RM systems (Table 4).

//=====

11. Table 4 would be more informative if added a column showing whether PacBio methylation observed is in line with prediction.

//=====

As per the reviewer's comment, we have added a new column "Motif identification" in Table 4.

//=====

12. Line 235: HCAGCTKC is actually not overlapping with GMAGCTKC, since H cannot be G.

//=====

Thank you for pointing out this. We have reflected this comment at L253 and L263 as below:

"overlapping" -> "similar"

//=====

13. Semantically, 'draft genome' could be used instead of 'genome bin'.

//=====

We have incorporated the reviewer's comment by rephrasing "genome bin" by "draft genome" or simply "genome" for the entire manuscript. We have also revised Table 2, 3, 4, and S3.

//=====

In addition, instead of random letters and numbers, throughout the manuscript it is better to keep the name of the genome at least in the Phylum level (e.g. Proteobacteria1).

//=====

Accordingly, to improve the readability, we have added phylum names to the genome bin IDs (e.g.,

“Proteobacteria BS1”) through the manuscript. Because the bin IDs are used for naming CDSs and correspondence information between bins and CDSs (*e.g.*, EMGBS15_03820 MTase belonged to BS15 genome) are is also important, we are keeping the ID numbers.

//=====

14. Unclear sentences, like line 61 “ Freshwater habitats are rich in phage-prokaryotic interactions, which are known to be closely related to prokaryotic DNA methylation...”.

//=====

We have revised the sentence as below:

L60: In addition, freshwater habitats are rich in phage-prokaryote interactions²⁴⁻²⁷, which can affect prokaryotic DNA methylation. We report that our CCS analyses of the environmental microbial samples allowed reconstruction of draft genomes and the identification of their methylated motifs, at least 9 of which were novel.

//=====

15. It is not accurate that the E.coli HST04 strain lacks endogenous MTase. It is only deleted for the *dam* and *dcm* genes, but still retains two other MTases (<http://rebase.neb.com/cgi-bin/acroegget?EcoHST04+o24091>)

//=====

We thank the reviewer very much for pointing out this important point. We have revised the sentences as follows:

L156: “The constructs were transformed into Escherichia coli HST04 *dam*⁻/*dcm*⁻ (Takara Bio), which lacks *dam* and *dcm* MTase genes.”

L337: deleted the sentence “that lacked endogenous MTases”

//=====

16. Line 85 –“...slight modifications” – what are the modifications?

19. Line 84 – Microbes were collected on the Sterivex filters, not DNA.

//=====

The modifications here were what described just below that sentence. In addition, what Sterivex filters capture is actually not DNA (as mentioned in the previous subsection). We have revised as below:

L86: The microbial DNA was retrieved using a PowerSoil DNA Isolation Kit (QIAGEN) according to the supplier’s protocol with slight modifications as described below.

//=====

17. Line 169: should be “reads” and line 179 should be “subreads”.

//=====

Regarding Line 169, what are presented are numbers of subreads, not those of reads, each of which contained multiple subreads. Here, numbers of subreads should be more useful information to readers, because reads also contain adapter sequences and what matters the analysis quality is the amount of subreads. Regarding Line 179, we could not identify what phrase is proposed to be replaced; however, “CCS reads” appearing around that line should also remain, as they represent CCS reads after consensus generation. To clarify what those words and phrases mean, we have revised the Method section as follows:

L100: Reads that contained at least three full-pass subreads on each polymerase read were retained to generate circle consensus sequence reads (CCS reads) using the standard PacBio SMRT software package with the default settings.

//=====

18. Methods – how was size selection performed? Also, how were SMRT sequencing libraries created. More experimental detail would be required for a reader to know how to reproduce these results.

20. Lines 89-90 – how were CCS libraries prepared? Most readers will not be familiar with this method, so additional details as well as the purpose of this type of library creation will be important for the general audience.

21. I would not consider running the same library on 2 SMRT cells to really be a “technical replicate”.

//=====

According to the reviewer’s suggestions, we have added details of the library construction steps as follows. In addition, we have removed the phrase “technical replicate”.

L91: SMRT libraries for CCS were prepared with a 4-kbp insertion length and two SMRT cells were used for each sample. Briefly, 3-5 kbp DNA fragments from each genomic DNA sample were extracted using the BluePippin size-selection system (Sage Science). Two sequencing libraries for CCS analysis were prepared using the SMRTbell Template Prep Kit 1.0-SPv3 according to the manufacturer’s protocol (Pacific Biosciences). The final libraries were sequenced using a PacBio Sequel sequencer with Sequel SMRT Cell 1M v2 and Sequel Binding/Sequencing Kits 2.0.

//=====

22. Figure 1 – reads should be corrected for read size in this figure. If this has already been done, please clarify in the figure legend.

//=====

In Figure 1, we are presenting a rough estimation of relative abundances of different phylogenetic clades. We agree that it may be better to do normalization by considering “genome size” (instead of read size), because microbes with large genomes have more DNA and likely appear more frequently in sequence data than expected by their cell numbers; however, genome size data are not available for all reads. On the other hand, corrections using read size are not expected to essentially change the figure, because after DNA extraction DNA molecules are sequenced regardless of their phylogenetic origin and/or genome size. Read lengths change by chance, but will rather introduce variance in the abundance estimation; thus, we would like the current figure that is based on the CCS read numbers.

//=====

23. Figure 2 – what is the size cutoff for inclusion in this figure?

//=====

All contigs were used for making this figure (i.e., there was no size cutoff). To make it clear that the bubble size reflects the sequence length of each assembled contig, we have revised the legend title of Figure 2:

“Size (bp)” -> “Contig length (bp)”

//=====

24. Figure S7 – There is inadequate detail in the figure legend or manuscript for me to interpret this figure.

//=====

We have removed Figure S7 at this revision (see our reply to Comment 4).

#####

Additional collections

#####

During this revision, we have found some minor errors in our manuscript. We apologize for them and have revised them as below:

L47: “methylation-detection errors” -> “modification detection errors”

L115: “binned genome” -> “assembled contigs”

L129: “methylation” -> “modification”

L173: “Data deposition” -> “Data availability”

L222: Among a total of 554 and 345 contigs, 290 (52.3%) and 100 (29.0%) were assigned to fifteen and four bins from the shallow and deep samples, respectively.

L337: “~, which we then transformed forced their expression, ~” -> “~, transformed them to E. coli cells, forced their expression ~”

Table S4: changed label names: “ (blank) ”->”Sample”

REVIEWERS' COMMENTS:

Reviewer #1 (Remarks to the Author):

The authors have responded well to my comments. I can accept the current version.

Reviewer #2 (Remarks to the Author):

Overall, the manuscript is much improved. I appreciate the authors' careful attention to detail in responding to the comments.

I have a few remaining concerns:

Major

1) What is the evidence that CCS improves methylation calling accuracy?

Minor

1) Would remove the word "severely" in line 20

2) line 22 - would change the word "must" to "likely"

3) Line 24 - what number of draft genomes arose from organisms that have yet to be cultured? A quantitative report here would be more informative than "most", as written.

4) Line 29 - would avoid use of the term "prove"

5) This is a small point, but when the authors indicate that samples were kept "cool in the dark" on line 80, do they know what the actual temperature was? If yes, please do report.

6) Line 113/line 214 - how many reads were removed (i.e. what proportion of total reads) due to repetitive contigs?

7) Line 291 - 292 - sentence wording is confusing.

8) Where are the data that support the statement in line 314-315, "The two MTases showed the best sequence similarities to those with m6A and m5C methylation activities...".

9) Line 332 - 333 - sentence wording is slightly confusing.

10) I remain unenthusiastic about the representation of proportion of reads in Figure 1 - but understand the author's perspective.

11) For Figure 2, it would be nice if the graph had a white background instead of a gray background (which makes it difficult for me to see some of the smaller, light colored dots).

#####

Reviewer #1 (Remarks to the Author):

#####

//=====

The authors have responded well to my comments. I can accept the current version.

//=====

Again, we would like to express our warm appreciation to the reviewer for the insightful comments on our manuscript.

#####

Reviewer #2 (Remarks to the Author):

#####

//=====

Overall, the manuscript is much improved. I appreciate the authors' careful attention to detail in responding to the comments.

1) What is the evidence that CCS improves methylation calling accuracy?

//=====

The methylation calling accuracy is improved when the PacBio sequencing depth is increased, and CCS increases the sequencing (subread) depth in expense of coverage (completeness in this context). Thus, in principle, CCS should improve the methylation calling accuracy, but, to be honest, we do not have any experimental evidence. In particular, because we adopted a depth threshold in this study, continuous long read (CLR) sequencing might also have led to accurate methylation call for the regions where the sequence depth exceeded the threshold. On the other hand, the methylation calling accuracy also depends on reference sequence accuracy. Here, it is established that CCS improve genomic sequencing and assembly accuracies for metagenomic samples (Frank et al., Sci. Rep. 2016). Therefore, we are confident that CCS improves the methylation calling accuracy in a metaepigenomic analysis in total; however, at this revision, we have removed "to achieve high accuracy" from Introduction to carefully avoid giving an impression that we have direct evidence on this issue.

L53-54: therefore, a cultivated clonal population is not required.

//=====

3) Line 24 - what number of draft genomes arose from organisms that have yet to be cultured? A

quantitative report here would be more informative than “most”, as written.

//=====

We deeply understand the rationale behind this question and actually wanted to provide such information in this context. However, a fundamental problem was that it heavily depends on the definitions of “organism” and “cultured”. Regarding the former, if we ignore differences within the genus level, BS7, BS12, and BS14 might be the same as cultured groups (Ca. Planktophila, Ca. Methylopumilus/LD28, and Ca. Fonsibacter, respectively), but we do not have evidence (and belief) that they are actually the same “organisms”. Regarding the latter, all of those three cultured groups have not been described (i.e., no type strain is deposited); that means, researchers cannot analyze methylation systems of those microbial groups by obtaining cultured strains anyway. Thus, the number could be zero, three, or else, but we think that providing such an ambiguous number is not so informative here. Because the point is that not many (possibly no) culture is available for doing experiments on the microbes analyzed in this study and (as an additional serious issue) Abstract has the word size limitation, we would like to keep “most”.

//=====

5) This is a small point, but when the authors indicate that samples were kept “cool in the dark” on line 80, do they know what the actual temperature was? If yes, please do report.

//=====

Unfortunately, temperature data during the transportation are unavailable. The water bottles were kept cool by direct contact with ice packs in a cool box, and the ice packs had not melted until the arrival at the laboratory. We have revised the Method section as follows:

L315: The water samples were transferred to sterile bottles, kept cool by contact with ice packs in a dark cool box, and immediately transported to the laboratory.

//=====

6) Line 113/line 214 - how many reads were removed (i.e. what proportion of total reads) due to repetitive contigs?

//=====

We have added the read-number and proportion information accordingly:

L109: The CCS reads from the shallow and deep samples were assembled into 599 and 429 contigs, respectively, using Canu¹⁸. After removing 45 (7.5%) and 84 (19.6%) repetitive contigs, we retrieved 554 and 345 contigs, respectively (Supplementary Table 3).

//=====

8) Where are the data that support the statement in line 314-315, “The two MTases showed the best

sequence similarities to those with m6A and m5C methylation activities...”.

//=====

The data are in Table 4. Because the previous and next paragraphs also refer to the same Tables 3 and 4, we would like to avoid putting an in-line reference to the table here (otherwise, we will need to put so many references around). Instead, we have added the CDS IDs to the text for clarification:

L215: The two MTases EMGBD2_08760 and EMGBD2_08790 showed the best sequence similarities to those with m5C and m6A methylation activities, respectively, while the detected motif contained an m6A site.

//=====

10) I remain unenthusiastic about the representation of proportion of reads in Figure 1 - but understand the author's perspective.

//=====

We have revisited Figure 1, but decided to keep it. We would like to thank the reviewer for the careful consideration.

//=====

- 1) Would remove the word “severely” in line 20
- 2) line 22 - would change the word “must” to “likely”

//=====

Due to the revision of Abstract according to the request by Editor, these words have been removed.

//=====

- 4) Line 29 - would avoid use of the term “prove”
- 7) Line 291 - 292 - sentence wording is confusing.
- 9) Line 332 - 333 - sentence wording is slightly confusing.
- 11) For Figure 2, it would be nice if the graph had a white background instead of a gray background (which makes it difficult for me to see some of the smaller, light colored dots).

//=====

We have revised the texts and Figure 2 accordingly. We would like to thank the reviewer very much again for the careful and thoughtful comments.

L29: prove -> highlights

L186: Among the 20 detected MTases, 13 MTases did not show sequence similarities to MTases that recognize the motifs identified in our metaepigenomic analysis (Tables 3 and 4).

L227: Among the MTases whose sequences showed the best similarities to MTases that

recognize motifs incongruent with our metaepigenomic results,